# ENERGY-BASED MODELS FOR CONTINUAL LEARNING

## ABSTRACT

We motivate Energy-Based Models (EBMs) as a promising model class for continual learning problems. Instead of tackling continual learning via the use of external memory, growing models, or regularization, EBMs have a natural way to support a dynamically-growing number of tasks and classes and less interference with old tasks. We show that EBMs are adaptable to a more general continual learning setting where the data distribution changes without the notion of explicitly delineated tasks. We also find that EBMs outperform the baseline methods by a large margin on several continual learning benchmarks. These observations point towards EBMs as a class of models naturally inclined towards the continual learning regime.

## 1 INTRODUCTION

Humans are able to rapidly learn new skills and continuously integrate them with prior knowledge. The field of Continual Learning (CL) seeks to build artificial agents with the same capabilities (Parisi et al., 2019). In recent years, CL has seen increased attention, particularly in the context of classification problems. A crucial characteristic of continual learning is the ability to learn new data without forgetting prior data. Models must also be able to incrementally learn new skills, without necessarily having a notion of an explicit task identity. However, standard neural networks (He et al., 2016; Simonyan & Zisserman, 2014; Szegedy et al., 2015) experience the catastrophic forgetting problem and perform poorly in this setting. Different approaches have been proposed to mitigate catastrophic forgetting, but many rely on the usage of external memory (Lopez-Paz & Ranzato, 2017; Li & Hoiem, 2017), additional models (Shin et al., 2017), or auxiliary objectives and regularization (Kirkpatrick et al., 2017; Schwarz et al., 2018; Zenke et al., 2017; Maltoni & Lomonaco, 2019), which can constrain the wide applicability of these methods.

In this work, we focus on classification tasks. These tasks are usually tackled by utilizing normalized probability distribution (i.e., softmax output layer) and trained with a cross-entropy objective. In this paper, we argue that by viewing classification from the lens of training an un-normalized probability distribution, we can significantly improve continual learning performance in classification problems. In particular, we interpret classification as learning an Energy-Based Model (EBM) across seperate classes (Grathwohl et al., 2019). Training becomes a wake-sleep process, where the energy of an input data and its ground truth label is decreased while the energy of the input and another selected class is increased. This offers freedom to choose what classes to update in the CL process. By contrast, the cross entropy objective reduces the likelihood of *all* negative classes when given a new input, creating updates that lead to forgetting.

The energy function, which maps a data and class pair to a scalar energy, also provides a way for the model to select and filter portions of the input that are relevant towards the classification on hand. We show that this enables EBMs training updates for new data to interfere less with previous data. In particular, our formulation of the energy function allows us to compute the energy of a data by learning a conditional gain based on the input label which serves as an attention filter to select the most relevant information. In the event of a new class, a new conditional gain can be learned.

These unique benefits are applicable across a range of continual learning tasks. Most existing works (Kirkpatrick et al., 2017; Zhao et al., 2020) toward continual learning typically learn a sequence of distinct tasks with clear task boundaries (*Boundary-Aware*). Many of these methods depend on knowing the task boundaries that can provide proper moments to consolidate knowledge. However, this scenario is not very common in the real world, and a more natural scenario is the *Boundary-*

*Agnostic* setting (Zeno et al., 2018; Rajasegaran et al., 2020), in which data gradually changes without a clear notion of task boundaries. This setting has also been used as a standard evaluation in the continual reinforcement learning (Al-Shedivat et al., 2017; Nagabandi et al., 2018). Many common CL methods are not applicable to the *Boundary-Agnostic* scenario as the task boundaries are unknown or undefined. In contrast, EBMs are readily applied to this setting without any modification and are able to support both *Boundary-Aware* and *Boundary-Agnostic* settings.

There are four primary contributions of our work. First, we introduce energy-based models for classification CL problems in both boundary-aware and boundary-agnostic regimes. Secondly, we use the standard contrastive divergence training procedure and show that it significantly reduces catastrophic forgetting. Thirdly, we propose to learn new conditional gains during the training process which makes EBMs parameter updates cause less interference with old data. Lastly, we show that in practice EBMs bring a significant improvement on four standard CL benchmarks, split MNIST, permuted MNIST, CIFAR-10, and CIFAR-100. These observations towards EBMs as a class of models naturally inclined towards the CL regime.

## 2 RELATED WORK

### 2.1 CONTINUAL LEARNING SETTINGS

**Boundary-aware versus boundary-agnostic**. In most existing continual learning studies, models are trained in a "boundary-aware" setting, in which a sequence of distinct tasks with clear task boundaries is given (e.g., Kirkpatrick et al., 2017; Zenke et al., 2017; Shin et al., 2017). Typically there are no overlapping classes between any two tasks; for example task 1 has data with ground truth class labels "1,2" and task 2 has data with ground truth class labels "3,4". In this setting, models are first trained on the entire first task and then move to the second one. Moreover, models are typically told when there is a transition from one task to the next. However, it could be argued that it is more realistic for tasks to change gradually and for models to not be explicitly informed about the task boundaries. Such a boundary-agnostic setting has been explored in (Zeno et al., 2018; Rajasegaran et al., 2020; Aljundi et al., 2019). In this setting, models learn in a streaming fashion and the data distributions gradually change over time. For example, the percentage of "1s" presented to the model might gradually decrease while the percentage of "2s" increases. Importantly, most existing continual learning approaches are not applicable to the boundary-agnostic setting as they require the task boundaries to decide when to consolidate the knowledge (Zeno et al., 2018). In this paper, we will show that our proposed approach can also be applied to the boundary-agnostic setting.

**Task-incremental versus class-incremental learning**. Another important distinction in continual learning is between task-incremental learning and class-incremental learning (van de Ven & Tolias, 2019; Prabhu et al., 2020). In task-incremental learning, also referred to as the multi-head setting (Farquhar & Gal, 2018), models have to predict the label of an input data by choosing only from the labels in the task where the data come from. On the other hand, in class-incremental learning, also referred to as the single-head setting, models have to chose between the classes from all tasks so far when asked to predict the label of an input data. Class-incremental learning is substantially more challenging than task-incremental learning, as it requires models to select the correct labels from the mixture of new and old classes. So far, only methods that store data or use replay have been shown to perform well in the class-incremental learning scenario (Rebuffi et al., 2017; Rajasegaran et al., 2019). In this paper, we try to tackle class-incremental learning without storing data and replay.

### 2.2 CONTINUAL LEARNING APPROACHES

In recent years, numerous methods have been proposed for CL. Here we broadly partition these methods into three categories: task-specific, regularization, and replay-based approaches.

**Task-specific methods.** One way to reduce interference between tasks is by using different parts of a neural network for different problems. For a fixed-size network, such specialization could be achieved by learning a separate mask for each task (Fernando et al., 2017; Serra et al., 2018), by *a priori* defining a different, random mask for every task to be learned (Masse et al., 2018) or by using a different set of parameters for each task (Zeng et al., 2019; Hu et al., 2019). Other methods let a neural network grow or recruit new resources when it encounters new tasks, examples of which are progressive neural networks (Rusu et al., 2016) and dynamically expandable networks (Yoon et al., 2017). Although these task-specific approaches are generally successful in reducing catastrophic forgetting, an important disadvantage is that they require knowledge of the task identity at both training and test time. These methods are therefore not suitable for class-incremental learning.

**Regularization-based methods.** Regularization is used in continual learning to encourage stability of those aspects of the network that are important for previously learned tasks. A popular strategy is to add a regularization loss to penalise changes to model parameters that are important for previous tasks. EWC (Kirkpatrick et al., 2017)) and online EWC (Schwarz et al., 2018) evaluate the importance of each parameter using the diagonal elements in the fisher information matrices, while SI (Zenke et al., 2017) tracks the past and current parameters and estimates their importance online. An alternative strategy is to regularize the network at the functional level. Learning without Forgetting (Li & Hoiem, 2017) uses knowledge distillation to encourage stability of the network's learned input-output mapping. However, these regularization-based approaches gradually reduce the model's capacity for learning new tasks and they have been shown to consistently fail in the class-incremental learning (Farquhar & Gal, 2018; van de Ven & Tolias, 2019).

**Replay methods.** To preserve knowledge, replay methods periodically rehearse previously acquired information during training (Robins, 1995). One way to do this, referred to as exact or experience replay, is to store data from previous tasks and revisit them when training on a new task. Although this might seem straightforward, critical non-trivial questions are how to select the data to be stored as well as exactly how to use them (Rebuffi et al., 2017; Lopez-Paz & Ranzato, 2017; Rajasegaran et al., 2019; Hou et al., 2019; Wu et al., 2019; Mundt et al., 2020). An alternative to storing data is to generate the data to be replayed. In such generative replay (Shin et al., 2017), a generative model is sequentially trained to generate input samples representative of those from previously seen tasks. While both types of replay can prevent catastrophic forgetting in both task- and class-incremental learning settings, an important disadvantage is that these methods are computationally relatively expensive as each replay event requires at least one forward and one backward pass through the model. In addition, storing data might not always be possible while incrementally training a generative model is a challenging problem in itself (Lesort et al., 2019; van de Ven et al., 2020).

In contrast, our EBMs reduce catastrophic forgetting without requiring knowledge of task-identity, without gradually restricting the model's learning capabilities and without using stored data.

## 3 BACKGROUND

In this section, we will introduce traditional continual learning methods, which are typically modified based on the feed-forward classifier. We will show the limitation of using such structures and how EBMs can be applied to these problems in the next section.

### 3.1 CONTINUAL LEARNING WITH FEED-FORWARD CLASSIFIER

Define $T$ as a classification task with inputs $\mathbf{x}$ and associated discrete classification labels $\mathbf{y} \in Y$, where $Y = \{1, \ldots, N\}$ contains $N$ classes. Traditional classifier approaches to continual learning typically predict the probabilities of all the $N$ classes using a feed-forward network

$$p_\theta(\mathbf{y}|\mathbf{x}) = \frac{\exp([f(\mathbf{x})]_{\mathbf{y}})}{\sum \exp([f(\mathbf{x})]_{\mathbf{y}_i})}, \quad \mathbf{y}_i \in Y \tag{1}$$

The input is a data $\mathbf{x} \in \mathbb{R}^D$ and the output is probabilities of all classes $p_\theta \in \mathbb{R}^N$. $f(\mathbf{x}) : \mathbb{R}^D \to \mathbb{R}^N$ is the feed-forward function that maps data $x$ into a $N$-dimensional vector. The final layer in such classifier networks is predefined to predict up to $N$ different classes.

Existing continual learning methods (Kirkpatrick et al., 2017; Masse et al., 2018) tackle the continual learning problem typically using a feed-forward classifier. However, such classifier architectures do not perform well in the *Class-IL* setting, where models have to predict the label of a given data from all classes (Tao et al., 2020a; He et al., 2018). The standard feed-forward classifier computes the Softmax and cross-entropy loss over all seen classes. When training on a new task, they improve the likelihood of ground truth classes but suppress the likelihood of old classes since they are all negative classes for data in the current task. This operation introduces competitive, winner-take-all dynamics that impact the strength of old classes and thus make the classifier forget past tasks.

### 3.2 ENERGY-BASED MODELS

Energy based models (LeCun et al., 2006) are a class of maximum likelihood models that define the likelihood of a data point $\mathbf{x} \in \mathbb{R}^D$ using the Boltzmann distribution

$$p_\theta(\mathbf{x}) = \frac{\exp(-E_\theta(\mathbf{x}))}{Z(\mathbf{x})}, \quad Z(\mathbf{x}) = \int_{\mathbf{x}} \exp(-E_\theta(\mathbf{x})) \tag{2}$$

where $E_\theta(\mathbf{x}) : \mathbb{R}^D \to \mathbb{R}$, known as the energy function, maps each input data to a scalar, and $Z(\mathbf{x})$ is the partition function over all data points.

EBMs are powerful generative models that have been applied to different tasks, such as structured prediction (Belanger & McCallum, 2016; Gygli et al., 2017; Rooshenas et al., 2019; Tu & Gimpel, 2019), machine translation (Tu et al., 2020), and image generation (Xie et al., 2016; 2018; Grathwohl et al., 2019). (Belanger & McCallum, 2016) introduce energy networks for structured prediction. They propose energy networks can be naturally posed as structured prediction, since the labels exhibit rich interaction structure. Tu & Gimpel (2019) propose a structured energy function for sequence labeling tasks. Rooshenas et al. (2019) introduce a search-guided approach to obtain training pairs using a combination of the energy function and reward function that allows the lightly-supervised training of structured prediction energy networks. Tu et al. (2020) treat the non-autoregressive translation system as an inference network that finds the translation by minimizing the energy of a pretrained autoregressive model. Xie et al. (2016) is the first paper to use ConvNet-parameterized EBMs with Langevin dynamics for image generation. They show that a generative random field model can be derived from the commonly used discriminative ConvNet. (Xie et al., 2018) cooperatively train the descriptor and generator using MCMC sampling.

## 4 Continual Learning with Energy-Based Models

In this section, we describe how EBMs can be used for classification and we discuss how they can overcome the above mentioned problems in the continual learning setting. Note our analysis is based on the *Class-IL* setting which is one of the most natural and also the hardest settings used in CL (van de Ven & Tolias, 2019; He et al., 2018; Tao et al., 2020b).

### 4.1 Energy-based models for classification

Different from existing works using EBM for structured prediction (Belanger & McCallum, 2016; Gygli et al., 2017; Rooshenas et al., 2019), such as multi-label classification tasks, CL does not access previous data while training on new data and thus the biggest challenge is to prevent catastrophic forgetting. The main focus of this paper is to mitigate the catastrophic forgetting in CL without using replay buffers to store past data or models.

To make the EBM suitable for classification, we use the Boltzmann distribution to define the likelihood of $\mathbf{y}$ conditioned on $\mathbf{x}$:

$$p_\theta(\mathbf{y}|\mathbf{x}) = \frac{\exp(-E_\theta(\mathbf{x}, \mathbf{y}))}{Z(\mathbf{y}|\mathbf{x})}, \quad Z(\mathbf{y}|\mathbf{x}) = \sum_{\mathbf{y}_i} \exp(-E_\theta(\mathbf{x}, \mathbf{y}_i)) \tag{3}$$

where $E_\theta(\mathbf{x}, \mathbf{y}) : (\mathbb{R}^D, \mathbb{N}) \to \mathbb{R}$ is the energy function that maps $\mathbf{x}$ and $\mathbf{y}$ to a scalar energy value, and $Z(\mathbf{y}|\mathbf{x})$ is the partition function that ensures $\sum_{\mathbf{y}} p_\theta(\mathbf{y}|\mathbf{x}) = 1$ for any given $\mathbf{x}$.

To minimize the negative log likelihood of the ground truth label $\mathbf{y}^+$, inspired by the contrastive divergence proposed by (Hinton, 2002), we minimize the energy of $\mathbf{x}$ at the ground truth label $\mathbf{y}^+$ while increasing the energy of $\mathbf{x}$ at other labels. We use the following loss and its corresponding gradient (see Appendix B for the derivation):

$$\mathcal{L}_{\text{ML}}(\theta) = \mathbb{E}_{\mathbf{x} \sim T}[E_\theta(\mathbf{x}, \mathbf{y}^+) + \log \sum_{\mathbf{y}_i \in \{\mathcal{N}, \mathbf{y}^+\}} \exp(-E_\theta(\mathbf{x}, \mathbf{y}_i))]. \tag{4}$$

$$\frac{\partial \mathcal{L}_{\text{ML}}}{\partial \theta} = \mathbb{E}_{\mathbf{x} \sim T} \left( \frac{\partial E_\theta(\mathbf{x}, \mathbf{y}^+)}{\partial \theta} - \mathbb{E}_{\mathbf{y}_i \sim \{\mathcal{N}, \mathbf{y}^+\}} \left[ \frac{\partial E_\theta(\mathbf{x}, \mathbf{y}_i)}{\partial \theta} \right] \right). \tag{5}$$

whereby $\mathcal{N}$ is the set of 'negative classes'. Following common practice in contrastive divergence, we sample the set of negative classes $\mathcal{N}$ from the set of class labels in the current batch $Y_B$, using only a single negative class per training sample. We note however that it is possible to use different strategies for choosing the negative classes, and in the experiments reported in the top half of Table 2 we explore alternative strategies. If there are batches that contain only a single class label, sampling a negative class from the current batch does not work, but one could instead select whichever other class was seen most recently as a negative class, or one could sample from all other classes seen so far. Another solution might be to always use a fake label representing "not exist" as negative sample.

As opposed to the classifier architectures that output a fixed $N$-dimensional probability vector of all classes, this formulation thus provides freedom in the choice of which class labels to train on.

This is important because an important issue with standard classifier models is that they suppress the strength of old classes and thus make the classifier forget past tasks as mentioned in Section 3.1. In contrast, EBMs define the probability without normalizing over all classes but instead sample a limited number of negative classes. This operation cause less interference with old classes without using any extra information which contributes to why EBMs suffer less from catastrophic forgetting.

Another advantage of our EBMs is that the choice of model architectures also becomes more flexible. The traditional classification models can only feed in $\mathbf{x}$ as input and the classification results only depend on the neural network trained on the current data $\mathbf{x}$. Thus the models usually fail on the old data after updating network parameters trained on the new data. In contrast, EBMs have many different ways to combine $\mathbf{x}$ and $\mathbf{y}$ in the energy function $E_\theta(\mathbf{x}, \mathbf{y})$ with the only requirement that $E_\theta(\mathbf{x}, \mathbf{y}) : (\mathbb{R}^D, \mathbb{N}) \to \mathbb{R}$. In EBMs, we can treat $\mathbf{y}$ as an attention filter or gate to select the most relevant information between $\mathbf{x}$ and $\mathbf{y}$ as described in Section 4.2.

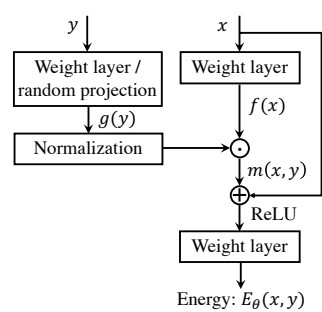

Figure 1: Model architecture used in EBM. EBM takes a data $\mathbf{x}$ and a class $\mathbf{y}$ as input and outputs their energy value.

Since the negative sample can be sampled from the current batch, EBMs do not require knowledge of the task on hand, allowing applications across different continual learning scenarios, including those when the underlying task is unclear. More broadly speaking, EBMs can go out of the scope of classification problems, and similar ideas can be applied to regression (Gustafsson et al., 2020), generation (Xie et al., 2016; Du et al., 2020), and reinforcement learning tasks (Parshakova et al., 2019), although we do not explore them in this work.

## 4.2 ENERGY NETWORK.

To compute the energy of any data $\mathbf{x}$ and class label $\mathbf{y}$ pair, we use $\mathbf{y}$ to influence a conditional gain on $\mathbf{x}$, which serves as an attention filter (Xu et al., 2015) to select the most relevant information between $\mathbf{x}$ and $\mathbf{y}$. In Figure 1, we first send $\mathbf{x}$ into a small network to generate the feature $f(\mathbf{x})$. The label $\mathbf{y}$ is mapped into a same dimension feature space $g(\mathbf{y})$ using a FC layer or a random projection. We use the Softmax function over each channel of $g(\mathbf{y})$ and combine it with $\mathbf{x}$:

$$m_c(\mathbf{x}, \mathbf{y}) = f_c(\mathbf{x}) \cdot \frac{\exp(g_c(\mathbf{y}))}{\sum_j \exp(g_j(\mathbf{y}))}, \tag{6}$$

where $f_c$ and $g_c$ are the $c^{\text{th}}$ channel of $f(\mathbf{x})$ and $g(\mathbf{y})$ respectively. The output is finally sent to a fully connected layer to generate the energy value $E_\theta(\mathbf{x}, \mathbf{y})$. For more details on our model architectures, see Appendix C.

Our EBMs allow any number of classes in new batches by simply training or defining a new conditional gain $g(\mathbf{y})$ for the new classes and generating its energy value with data point $\mathbf{x}$. This formulation gives us freedom to learn new classes without pre-defining their number in advance. We note that also in a standard classifier new classes can be dynamically added, but there new class heads need to be added to the softmax output layer.

**Class inference.** During inference, the model must predict a class label from all classes seen so far. let $\mathbf{x}_k$ be one data point from a batch $B_k$ with an associated discrete label $\mathbf{y} \in Y_k$, where $Y_k$ contains classes in $B_k$. There are $Y = \bigcup_{k=1}^K Y_k$ different classes in total after seeing all the batches. The MAP estimate is

$$\hat{\mathbf{y}} = \arg\min_{\mathbf{y}} E_{\theta_K}(\mathbf{x}_k, \mathbf{y}), \quad \mathbf{y} \in \bigcup Y_k, \tag{7}$$

where $E_{\theta_K}(\mathbf{x}_k, \mathbf{y})$ is the energy function with parameters $\theta_K$ resulting from training on the batches $\{B_1, \cdots, B_K\}$. The energy function can compute an energy for any discrete class input, including unseen classes. This avoids needing to predefine the number of classes in advance, as is necessary with traditional CL models.

## 5 EXPERIMENTS

In this section, we want to investigate several questions, including how does the proposed EBMs perform on different CL settings? Can we qualitatively understand the differences between EBMs and baselines? Is the proposed EBM training objective better than other objectives? Can we apply the EBM training objective to other baseline methods? And finally what is the the best architecture for

Table 1: Evaluation of the *Class-IL* performance on the *Boundary-Aware* setting compared to different baselines. The test accuracy on four datasets are reported. Each experiment is performed 20 times with different random seeds, the results are reported as the mean ± SEM over these runs. Note both EBMs and baselines do not use extra memory to store data or models for fair comparison.

| Method | splitMNIST | permMNIST | Cifar10 | Cifar100 |
|--------|-----------|-----------|---------|----------|
| CLS | $19.91 \pm 0.01$ | $16.95 \pm 0.31$ | $19.06 \pm 0.01$ | $8.18 \pm 0.10$ |
| EWC | $20.03 \pm 0.09$ | $25.14 \pm 0.71$ | $18.99 \pm 0.01$ | $8.20 \pm 0.09$ |
| Online | $19.89 \pm 0.02$ | $33.20 \pm 0.74$ | $19.07 \pm 0.01$ | $8.38 \pm 0.15$ |
| SI | $19.98 \pm 0.03$ | $29.35 \pm 0.90$ | $19.14 \pm 0.02$ | $9.24 \pm 0.22$ |
| LwF | $20.53 \pm 0.11$ | $43.88 \pm 0.88$ | $19.20 \pm 0.30$ | $10.71 \pm 0.11$ |
| MAS | $19.50 \pm 0.30$ | - | $20.25 \pm 1.54$ | $8.44 \pm 0.27$ |
| BGD | $19.64 \pm 0.03$ | $84.78 \pm 1.30$ | - | - |
| **EBM** | $\mathbf{53.12 \pm 0.04}$ | $\mathbf{87.78 \pm 0.01}$ | $\mathbf{40.19 \pm 0.01}$ | $\mathbf{30.28 \pm 0.28}$ |

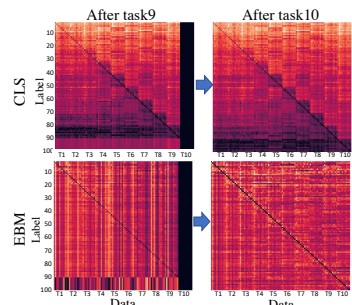

Figure 2: Energy landmaps of CLS and EBMs after training on task $T_9$ and $T_{10}$ on permuted MNIST. The darker the diagonal is, the better the model is in preventing forgetting previous tasks.

label conditioning that causes the minimum interference with old data? To answer these questions, we first introduce experiments on the *Boundary-Aware* setting which typically learn a sequence of distinct tasks with clear task boundaries in Section 5.1. We then show that EBMs can be naturally applied to the *Boundary-Agnostic* setting where the data gradually changing in a steaming fashion without the notion of task boundaries in Section 5.2.

## 5.1 EXPERIMENTS ON BOUNDARY-AWARE SETTING

### 5.1.1 DATASETS AND EVALUTION PROTOCOLS

**Datasets.** We evaluate the proposed EBMs on the split MNIST (Zenke et al., 2017), permuted MNIST (Kirkpatrick et al., 2017), CIFAR-10 (Krizhevsky et al., 2009), and CIFAR-100 (Krizhevsky et al., 2009) datasets. We follow existing approaches and separate the datasets into several splits. The split MNIST dataset is obtained by splitting the original MNIST dataset (LeCun et al., 1998) into 5 tasks with each task has 2 classes. It has 60,000 training images and 10,000 test images. The permuted MNIST has 10 tasks, each task with 10 classes. For each task, the original images pixels are randomly permuted to generate $32 \times 32$ images. We separate CIFAR-10 into 5 tasks, each task with 2 classes. There are 50,000 training images and 10,000 test images. Similarly, CIFAR-100 is split into ten tasks with each task has 10 classes.

**Evaluation protocols.** Task-incremental Learning (*Task-IL*), Domain-incremental Learning (*Domain-IL*), and Class-incremental Learning (*Class-IL*) are three common evaluation metrics used by existing works (van de Ven & Tolias, 2019; Prabhu et al., 2020). Most approaches perform fairly good on the first two simpler settings, but fail on the *Class-IL* setting, which is considered as the most natural and also the hardest setting for continual learning (Tao et al., 2020a; He et al., 2018; Tao et al., 2020b). *Class-IL* predicts the label of a given data from all seen classes. In this paper, we consider *Class-IL* as our evaluation metric.

### 5.1.2 COMPARISONS WITH EXISTING METHODS

Due to the difficulty on the class incremental setting, most approaches rely an external quota of memory. Such replay based methods (Shin et al., 2017; Rebuffi et al., 2017) use extra memory, larger or extra models, and are computationally relatively expensive. In this paper, we focus on continual learning without using replay and without storing data.

We compare the proposed method with available baseline models that do not use replay or stored data, including a standard classification model (CLS), EWC (Kirkpatrick et al., 2017), Online EWC (Schwarz et al., 2018), SI (Zenke et al., 2017), LwF (Li & Hoiem, 2017), MAS (Aljundi et al., 2019), and BGD (Zeno et al., 2018). The continual learning results on four datasets are shown in Table 1. All the baselines and EBMs are based on the same model architecture. For split MNIST and permuted MNIST, we use several fully-connected layers as in (van de Ven & Tolias, 2019). For CIFAR-10 and CIFAR-100, we use the convolutional network (see Appendix C). For all the baselines and EBMs on CIFAR-100, we pre-train the convolutional layers on CIFAR-10 and only fine-tune the fully-connected layers.

Similar training regimes are used for the EBMs and baselines. On the split MNIST, permuted MNIST, and CIFAR-10 datasets, we train for 2000 iterations per task. On the CIFAR-100 dataset we train for 5000 iterations per task. For all experiments, we use the adam optimizer with learning rate $1e^{-4}$. Each experiment in Table 1 is run 20 times with different random seeds, with results reported as the mean $\pm$ SEM. EBMs have a significant improvement on all the datasets, showing that EBMs forget less when updating models for new tasks.

### 5.1.3 QUALITATIVE ANALYSIS

**Energy landscape.** To better understand why EBMs suffer less from catastrophically forgetting, we qualitatively compare the change in energy landscapes of EBMs and baseline methods as the learning progresses. We show the energy landscapes after training on task 9 and task 10 on the permuted MNIST dataset in Figure 2. For the classifier, we show its negative probabilities on all classes. Each datapoint has 100 energy values (EBM) or probabilities (CLS) corresponding to the 100 labels in the dataset and the values are normalized over 100 classes for each datapoint. The dark diagonal elements indicate the model predictions are correct for both old and new data. After training on task $T_9$, CLS assigns high probabilities to classes from task $T_9$ (80-90) for almost all the data from task $T_1$ to $T_9$. However, the highest probabilities shift to classes from task $T_{10}$ (90-100) after learning task $T_{10}$. This means classifier tends to assign high probabilities to new classes for both old and new data, indicating forgetting. EBM on the other hand, tends to have low energies across the diagonal, which means that after training on new tasks, EBM still assigns low energies to true labels of previous data. This indicates that EBM is better at learning new tasks without catastrophically forgetting of old tasks.

**Predicted class distribution.** In Figure 3, we plot the proportional distribution of all seen classes so far on the splitMNIST dataset. Taking the second figure in the first row as an example, it shows the distribution of predicted labels after training on the second task which means there are four seen classes $\{1, 2, 3, 4\}$. For each image from the first and second task, we compute its top-1 class prediction and accumulate the prediction on each class to obtain the proportional distribution graph over all seen classes. Since the number of images in each class are almost the same, the ground truth proportional distribution should be

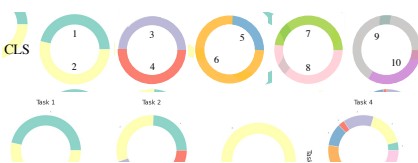

Figure 3: Predicted label distribution after learning each task on the split MNIST dataset. CLS only predicts classes from the current task while EBMs can predict classes for all seen classes.

uniform over all the classes. CLS can predict the first two classes uniformly after training on task 1 (first column). However, after learning new tasks, classes from the new task have larger proportions than old classes which means the model tends to predict new classes and forget the old classes. The last figure in row one gives very high proportions for classes from the last task but very low proportions for previously classes. CLS almost forgets all previous tasks and only predict classes from the last task even the input images are from old tasks. EBM covers all seen classes, indicating memorizing the old task and preventing forgetting.

We provide more analysis about EBMs and baselines from different perspectives in the Appendix, including the confusion matrix in Appendix A.2, the model capacity in Appendix A.3, and parameter importance in Appendix A.4.

### 5.1.4 IS THE STRONG PERFORMANCE OF EBMS DUE TO THE ENERGY TRAINING OBJECTIVE OR DUE TO THE LABEL CONDITIONING?

**Effect of energy training objective.** We conduct an experiment on the CIFAR-100 dataset to investigate how different training objectives influence the CL results. In Equation 4, the negative labels $\mathbf{y}_i$ ares sampled from $Y_B$. We test three different sampling strategies in Table 2. The first one uses all seen classes so far as negative labels which is similar to the traditional feed-forward classifiers (**All Neg Seen (V4)**). The second one takes all the classes in the current batch as negative labels (**All Neg Batch (V4)**). The last one randomly selects one class from the current batch as the negative as we reported in Section 4.1 (**1 Neg Batch (V4)**). Note that the negative labels do not include the ground truth class in all these three strategies. We found using only one negative sample generates the best result and using negatives sampled from classes in the current batch is better than from all seen classes. This conclusion is consistent with our analysis in Section 4.1. Since the loss in Equation 4 aims at improving the energy of negative samples while decreasing the energy of positive ones, sampling negatives from the current batch has less interference with previous classes than sampling

from all seen classes. Using a single negative causes the minimum suppression on negative samples and thus has the best result. Overall, our results indicate that surprisingly, and counterintuitively, directly optimizing the cross-entropy loss used by existing approaches may not be the best way for continual learning.

**Effect of label conditioning.** Next, we test whether the label conditioning in our EBMs is important for their performance. We test this by modifying the training objective of a standard classifier to that of our proposed energy objective, which is the same as running our EBMs without the label conditioning. The results of this comparison are listed in Table 3. The new training objective does not suppress the probability of old classes when improving the probability of new classes and thus achieves better results. However, EBMs still outperform the baselines, implying that the label conditioning architecture also contributes to why EBMs suffer less from catastrophic forgetting. We also show the testing accuracy curve in Appendix A.1.

To summarize, we showed that the strong performance of our EBMs is due to both the energy training objective and the label conditioning.

### 5.1.5 COMPARISON OF DIFFERENT ENERGY NETWORK ARCHITECTURES

EBMs allow flexibility in integrating data information and label information in the energy function. To investigate where and how to combine the information from $\mathbf{x}$ and $\mathbf{y}$, we conduct a series of experiments on the CIFAR-10 dataset. Table 2 shows four model architectures (V1-V4) that combine $\mathbf{x}$ and $\mathbf{y}$ in the early, middle, and late stages, respectively (see Appendix C for the architecture details). We find combining $\mathbf{x}$ and $\mathbf{y}$ in the late stage (V4) performs the best. We note that instead of learning a feature embedding of label $\mathbf{y}$, we can use a fixed projection matrix which is randomly sampled from the uniform distribution $\mathcal{U}(0, 1)$. Even using this fixed random projection can already generates better results than most baselines in Table 1. Note further that the number of trainable parameters in the "Fix" setting is much lower than that of the baselines. Using a learned feature embedding of $\mathbf{y}$ can further improve the performance. We may also apply different normalization methods over the feature channel of $\mathbf{y}$. We find that Softmax (**End Softmax (V4)**) is better than the L2 normalization (**End Norm2 (V4)**) and no normalization (**End (V4)**).

Table 2: Different EBM training objectives and label conditioning architectures.

| Dataset | Cifar100 |
|---|---|
| All Neg Seen (V4) | $8.07 \pm 0.10$ |
| All Neg Batch (V4) | $29.03 \pm 0.53$ |
| **1 Neg Batch (V4)** | $\mathbf{30.28 \pm 0.28}$ |

| Dataset | Cifar10 |
|---|---|
| Beginning (V1) | $13.69 \pm 0.02$ |
| Middle(V2) | $20.16 \pm 0.02$ |
| Middle(V3) | $18.36 \pm 0.02$ |
| End Fix (V4) | $34.30 \pm 0.03$ |
| End Fix Norm2 (V4) | $33.91 \pm 0.02$ |
| End Fix Softmax (V4) | $35.97 \pm 0.01$ |
| End (V4) | $38.13 \pm 0.01$ |
| End Norm2 (V4) | $37.23 \pm 0.02$ |
| **End Softmax (V4)** | $\mathbf{40.19 \pm 0.01}$ |

### 5.1.6 INTERFERENCE WITH PAST DATA

Here we test the effects of label conditioning and the energy training objective in a different way, by evaluating their ability to prevent interference with past data when learning classification on new classes. Formally, let $\mathbf{x}_i$ and $\mathbf{x}_j$ be two data points and $\theta_i$ be the model parameters after training on $\mathbf{x}_i$. The model parameters change $\triangle\theta(\mathbf{x}_j)$ after learning new data $\mathbf{x}_j$. We test the difference between the losses $\mathcal{L}_{\mathrm{ML}}(\mathbf{x}_i \mid \theta_i)$ and $\mathcal{L}_{\mathrm{ML}}(\mathbf{x}_i \mid \theta_i + \triangle\theta(\mathbf{x}_j))$. Ideally, we expect $\mathcal{L}_{\mathrm{ML}}(\mathbf{x}_i \mid \theta_i + \triangle\theta(\mathbf{x}_j)) \leq \mathcal{L}_{\mathrm{ML}}(\mathbf{x}_i \mid \theta_i)$ for $\mathbf{x}_i \neq \mathbf{x}_j$, which means the update on new data has no influence or positive influence on the old data. The first-order expansion gives $\mathcal{L}_{\mathrm{ML}}(\mathbf{x}_i \mid \theta_i + \triangle\theta(\mathbf{x}_j)) \approx \mathcal{L}_{\mathrm{ML}}(\mathbf{x}_i \mid \theta_i) + \nabla_{\theta_i}\mathcal{L}_{\mathrm{ML}}(\mathbf{x}_i \mid \theta_i)^T \triangle\theta(\mathbf{x}_j)$. To make our desired equality hold, the gradient at $\mathbf{x}_i$ should be $\nabla_{\theta_i}\mathcal{L}_{\mathrm{ML}}(\mathbf{x}_i \mid \theta_i)^T \triangle\theta(\mathbf{x}_j) \leq 0$. $\phi = \nabla_{\theta_i}\mathcal{L}_{\mathrm{ML}}(\mathbf{x}_i \mid \theta_i)^T \triangle\theta(\mathbf{x}_j)$ can be used to measure the influence of updating models on new data on the performance of old data. The smaller the value is, the less negative influence on old data, and thus the better the model is in preventing forgetting.

We compare EBMs with the standard classifier (CLS) and CLS using our EBM training objective (CLS*) (see Appendix A.5 for the implementation details). For results on split MNIST, the average value of $\phi$ over all tasks of CLS, CLS*, and EBMs are $\phi_{\mathrm{CLS}} = 0.168$, $\phi_{\mathrm{CLS*}} = 0.004$, and $\phi_{\mathrm{EBM}} = -3.743e^{-5}$, respectively. For CIFAR-10, the results are $\phi_{\mathrm{CLS}} = 1.104$, $\phi_{\mathrm{CLS*}} = 0.043$, and $\phi_{\mathrm{EBM}} = 4.414e^{-5}$, respectively. We found EBMs have smaller $\phi$ than CLS and CLS*, which is consistent with our analysis that the conditional gain $g(\mathbf{y})$ serves as an attention filter that could select the

Table 3: Results of baselines using our training objective and their original one.

| split MNIST | Ori | Ours |
|---|---|---|
| CLS | $19.90 \pm 0.02$ | $44.98 \pm 0.05$ |
| EWC | $20.01 \pm 0.06$ | $50.68 \pm 0.04$ |
| Online | $19.96 \pm 0.07$ | $50.99 \pm 0.03$ |
| SI | $19.99 \pm 0.06$ | $49.44 \pm 0.03$ |
| EBM | - | $\mathbf{53.12 \pm 0.04}$ |
| **CIFAR-10** | **Ori** | **Ours** |
| CLS | $19.06 \pm 0.01$ | $19.22 \pm 0.02$ |
| EWC | $18.99 \pm 0.01$ | $36.51 \pm 0.03$ |
| Online | $19.07 \pm 0.01$ | $36.16 \pm 0.02$ |
| SI | $19.14 \pm 0.02$ | $35.12 \pm 0.02$ |
| EBM | - | $\mathbf{40.19 \pm 0.01}$ |

Table 4: Evaluation of the *Class-IL* performance on the *Boundary-Agnostic* setting compared to different baselines. The test accuracy on four datasets are reported. Each experiment is performed 5 times with different random seeds, the results are reported as the mean $\pm$ SEM over these runs. Note both EBMs and baselines do not use extra memory to store data or models for fair comparison.

| Method | splitMNIST | permMNIST | Cifar10 | Cifar100 |
|---|---|---|---|---|
| CLS | $24.03 \pm 0.01$ | $21.42 \pm 0.01$ | $23.30 \pm 0.01$ | $9.85 \pm 0.02$ |
| Online | $39.62 \pm 0.04$ | $41.37 \pm 0.04$ | $22.53 \pm 0.01$ | $9.57 \pm 0.02$ |
| SI | $28.79 \pm 0.01$ | $35.71 \pm 0.01$ | $26.26 \pm 0.01$ | $10.42 \pm 0.01$ |
| BGD | $21.65 \pm 0.02$ | $26.15 \pm 0.02$ | $17.03 \pm 0.01$ | $8.50 \pm 0.02$ |
| **EBM** | $\mathbf{81.78 \pm 0.06}$ | $\mathbf{92.35 \pm 0.01}$ | $\mathbf{49.47 \pm 0.03}$ | $\mathbf{34.39 \pm 0.24}$ |

relevant information between $\mathbf{x}$ and $\mathbf{y}$ and make parameter updates cause less interference with old data, which contributes to why EBMs suffer less from catastrophic forgetting.

## 5.2 EXPERIMENTS ON BOUNDARY-AGNOSTIC SETTING

When applying continual learning in real life, boundaries are not usually well defined between different tasks. Most existing continual learning methods are often tailored to the problem setup where a fixed boundary between tasks is defined, and must be informed when the tasks are switched during training. We show that EBMs are able to flexibly perform continual learning across different problem setups, and perform well on the *Boundary-Agnostic* setting as well.

### 5.2.1 DATASETS AND EVALUTION PROTOCOLS

We evaluate our proposed continual learning method on the *Boundary-Agnostic* setting on four datasets, including split MNIST, permuted MNIST, CIFAR-10, and CIFAR-100. We use the code of "continuous task-agnostic learning" proposed by (Zeno et al., 2018) to generate the training/testing data. The transition between tasks goes slowly over time using a sampling function based on the probability of each task. There is a mixture of data from two different tasks during the transition and the proportion of new data gradually increases. Similar to the *Boundary-Aware* setting, We use *Class-IL* as our evaluation metric for the *Boundary-Agnostic* continual learning.

### 5.2.2 COMPARISON WITH EXISTING METHODS

We focus on the investigation on the performance of different model architectures with similar model size and memory footprint. We do not directly compare with replay-based methods as they use extra memory, larger or extra models. Since there is no knowledge on the number of tasks, previous methods of continual learning that rely on task boundaries are generally inapplicable. One trivial adaptation is to take the core action after every batch step instead of every task. However, doing such adaptation is impractical for most algorithms, such as EWC, because of the large computational complexity. We successfully get the result of CLS, Online EWC, SI, and BGD baselines. Overall classification results on four datasets are shown in Table 4.

All the baselines and EBM are based on the same model architecture as in the *Boundary-Aware* setting. Each experiment was performed 5 times with different random seeds, the results are reported as the mean $\pm$ SEM over these runs. We observe that EBMs have a significant improvement on all the datasets. EBMs can be naturally applied to both the *Boundary-Aware* and *Boundary-Agnostic* settings without any modification, as EBMs have a very flexible training objective. The experiments demonstrate EBMs have good generalization ability for different continual learning problems. EBMs can naturally handle different number of new classes, new tasks, data streams with and without clear task boundaries.

## 6 CONCLUSION

In this paper, we found that energy-based models are a promising class of models towards a variety of different continual learning settings. We demonstrated that EBMs exhibit many desirable characteristics to prevent catastrophic forgetting in CL, and we experimentally showed that EBMs obtain the state-of-the-art performance on the *Class-IL* and *Boundary-Agnostic* settings on multiple benchmarks.

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

## A    ADDITIONAL ANALYSES

Extending the results presented in Section 5.1, here we further compare EBMs with the baseline models by providing additional quantitative analyses of their performance. We show testing accuracy curves in Section A.1, confusion matrices between the ground truth labels and model predictions in Section A.2, model capacity comparisons in Section A.3, and parameter importance measurement in Section A.4. In Section A.5, we provide more details of the label conditioning analysis mentioned in Section 5.1.5. Finally, in Section A.6 we perform the split CIFAR-100 protocol with several different number of tasks to test the generality of the proposed EBMs.

### A.1    TESTING ACCURACY CURVE

In Section 5.1.4, we show that the proposed EBM training objective is also applicable to baseline approaches and improves their performance significantly on the *Class-IL* setting. Figure 4 shows the testing accuracy of each task as the training progresses. We compare the standard classifier (CLS), classifier using our training objective (CLS*), and our EBMs. The left figures show results on the split MNIST dataset while the right figures show results on the permuted MNIST dataset. We observe that the accuracy of old tasks in CLS drop sharply when learning new tasks, while the EBM training objective can mitigate the forgetting problem. The curve on EBMs drops even slower than CLS and CLS*, implying the proposed energy objective can mitigate the catastrophic forgetting problem.

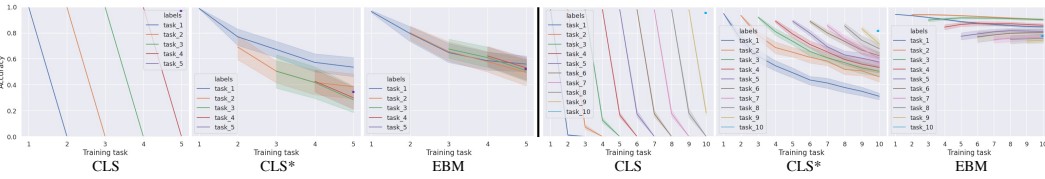

Figure 4: *Class-IL* testing accuracy of the standard classifier (CLS), classifier using our training objective (CLS*), and EBMs on each task on the split MNIST dataset (left) and permuted MNIST dataset (right).

### A.2    CLASS CONFUSION MATRIX AT THE END OF LEARNING

In Section 5.1.3, we show the qualitative analysis of EBMs and baselines. Except the energy landscape and predicted class distribution, we also generate the confusion matrices of EBMs and baseline approaches. Confusion matrix shows the relationship between ground truth labels and predicted labels. In Figure 5, we draw the confusion matrices after training on all the tasks on the split MNIST dataset and permuted MNIST dataset. The classifier tends to predict the classes from the last task (class 8, 9 for split MNIST and 90-100 for permuted MNIST). EBMs have high values along the diagonal that means the predicted results matching the ground truth labels for all the sequentially learned tasks. This demonstrates that EBMs are better at learning new tasks without catastrophically forgetting of old tasks.



Figure 5: Confusion matrices between ground truth labels and predicted labels at the end of learning on split MNIST (left) and permuted MNIST (right). The lighter the diagonal is, the more accurate the predictions are.

## A.3 MODEL CAPACITY

Another hypothesized reason for why EBMs suffer less from catastrophic forgetting than standard classifiers is potentially their larger effective capacity. To analyze effective capacity of our models, we test the model capacity of the standard classifier and EBMs on both the generated images and natural images.

**Model capacity on generated images.** We generate a large randomized dataset of $32 \times 32$ images with each pixel value uniformly sampled from -1 to 1. Each image is then assigned a random class label between 0 and 10. We measure the model capacity by evaluating to what extent the model can fit a such dataset. For both the standard classifier and the EBM, we evaluate three different sizes of models (small, medium, and large). For a fair comparison, we control the EBM and classifier have similar number of parameters. The Small EBM and CLS have $2, 348, 545$ and $2, 349, 032$ parameters respectively. The medium models have $5, 221, 377$ (EBM) and $5, 221, 352$ (CLS) parameters while the large models have $33, 468, 417$ (EBM) and $33, 465, 320$ (CLS) parameters. We use the model architectures in Table 5a and Table 5b for EBMs and classifiers.

The resulting training accuracies are shown in Figure 6 with the number of data ranges from one to five millions. Given any number of datapoints, EBM obtains higher accuracy than the classifier, demonstrating that indeed EBM has larger capacity to memorize data given a similar number of parameters. The gap between EBM and CLS increases when the models become larger. The larger capacity of EBM potentially enables it to memorize more data and mitigate the forgetting problem.

**Model capacity on natural images.** We also compare classifiers and EBMs on natural images from CIFAR-10. Each image is assigned a random class label between 0 and 10. We use the same network architecture as in Table 5a and Table 5b, but with a hidden unit size of $h = 256$. Since there are only $50, 000$ images on CIFAR-10, we use a small classifier and EBM and train them on the full dataset. After training 100000 iterations, the EBM obtains a top-1 prediction accuracy of $82.81$, while the classifier is $42.19$. We obtain the same conclusion that EBM has larger capacity to memorize data given a similar number of parameters.

Table 5: The model architectures used for the model capacity analysis. $h$ are 512, 1024, and 4096 for the small, medium and large network, respectively.

(a) The architecture of the EBM.

| x = FC(784, h) (x) |
| --- |
| x = ReLU (x) |
| y = Embedding (y) |
| x = x * y |
| x = ReLU (x) |
| out = FC(h, 1) (x) |

(b) The architecture of the standard classifier..

| x = FC(784, h) (x) |
| --- |
| x = ReLU (x) |
| x = FC(h, h) (x) |
| x = ReLU (x) |
| out = FC(h, 10) (x) |

## A.4 PARAMETER IMPORTANCE

To further understand why EBMs suffer less from catastrophic forgetting, we design an experiment to test the importance of model parameters on past data. Inspired by the elastic weight consolidation (EWC) (Kirkpatrick et al., 2017), we estimate the importance of parameters for each tasks using the diagonal elements of the Fisher information matrix (FIM) $F$. Let $\theta_i$ be the model parameters after training on task $T_i$. Given one of previous tasks $T_j, j < i$, we evaluate how important each parameter is for tasks $T_j$. The $k^{\text{th}}$ diagonal of $F$ is defined as the gradient on the EBM loss

$$F_{i,k} = \mathbb{E}_{\mathbf{x} \sim T_j} \nabla_{\theta_{i,k}} \left( E_{\theta_i}(\mathbf{x}, \mathbf{y}^+) + \log \sum_{\mathbf{y}'} \exp(-E_{\theta_i}(\mathbf{x}, \mathbf{y}')) \right)^2, \qquad (8)$$

where $\mathbf{x}$ is sampled from tasks $T_j$ and $E(\mathbf{x}, \mathbf{y}^+)$ is the energy value of the input data $\mathbf{x}$ and ground truth label $\mathbf{y}^+$. $\mathbf{y}' \in Y_j$ are classes randomly selected from the current batch. Here we use a single

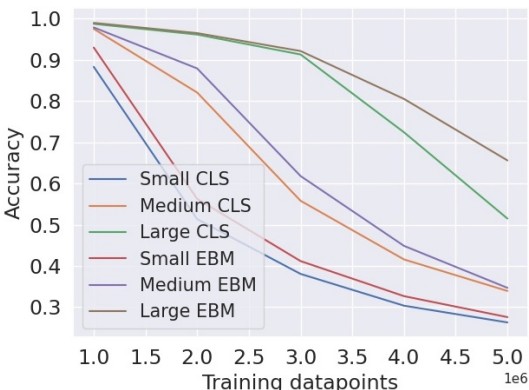

Figure 6: Model capacity of classifier and EBM using different model sizes.

negative class. The above equation assigns high values to parameters crucial to task $T_j$ as their gradients with respect to the loss are larger. Since the diagonal elements of the fisher information matrix measure the importance of each parameter to a given task, the density of diagonal elements represents the proportion of important parameters over all parameters. More density means more parameters are important for the given task and less parameters can be recruited for new tasks. Ideally, we expect these values to be sparse.

In Figure 7, we show the diagonal elements of the standard classifier (CLS), classifier using our training objective (CLS*), and our EBMs on the split MNIST dataset. For CLS and CLS*, we follow (Kirkpatrick et al., 2017) to compute their fisher information matrices. For comparisons across multiple models, we normalize the FIM diagonal elements of each method to be between 0 and 1 and report the normalized results in Figure 7. For example, "Fisher 5 on data 1" shows the diagonal elements of the Fisher information matrix obtained by Equation 8 using the model parameters $\theta_5$ (after training on task $T_5$) and data $\mathbf{x}, \mathbf{y}^+, \mathbf{y}'$ from task $T_1$. The distribution of EBMs is sparser than CLS and CLS* indicating that EBMs have fewer important parameters for previous data. Updating parameters for the new task will have less negative impact on old tasks. In addition, more parameters can be used for learning new tasks if the distribution is sparse. This may provide another explanation for why EBMs can mitigate catastrophic forgetting.

## A.5 MORE DETAILS OF THE INTERFERENCE WITH PAST DATA ANALYSIS

In Section 5.1.5, we test the interference of models trained on new data with past data. We compare the proposed EBMs with a standard classifier (CLS) and CLS using our EBMs training objective (CLS*). For CLS and CLS*, we use the cross-entropy loss to get their $\phi$, *i.e.,* the cross-entropy losses on the old data before and after training on a new data.

We obtain $\theta_i$ by training on 200 randomly selected data from task1 and compute its gradient after training. Then we perform a single step optimization on 200 data that are randomly selected from task2 and average their parameters to get $\theta_j$. We compute $\phi$ using the inner product of $\triangle\theta(\mathbf{x}_j) = \theta_j - \theta_i$ and the gradient $\nabla_{\theta_i}\mathcal{L}_{\text{ML}}(\mathbf{x}_i \mid \theta_i)$. For comparisons among multiple approaches, we normalize the $\phi$ of each method by dividing its maximum element value of $\theta_i$ and scale $\phi$ based on the length of $\theta$, as CLS and EBMs have different number of parameters.

## A.6 SPLIT CIFAR-100 WITH DIFFERENT NUMBERS OF TASKS

To test the generality of our proposed EBMs, in Table 6 we repeat the boundary-aware experiments on spit CIFAR-100 (see Table 1 in the main text) for different number of classes per task. In Table 1 the CIFAR-100 dataset was split up into 10 tasks, resulting in 10 classes per task. Here we additionally perform the split CIFAR-100 protocol with 5 different tasks (i.e., 20 classes per task), with 20 different tasks (i.e., 5 classes per task) and with 50 different tasks (i.e., 2 classes per task). For all settings, we find that our EBM substantially outperforms the baselines.

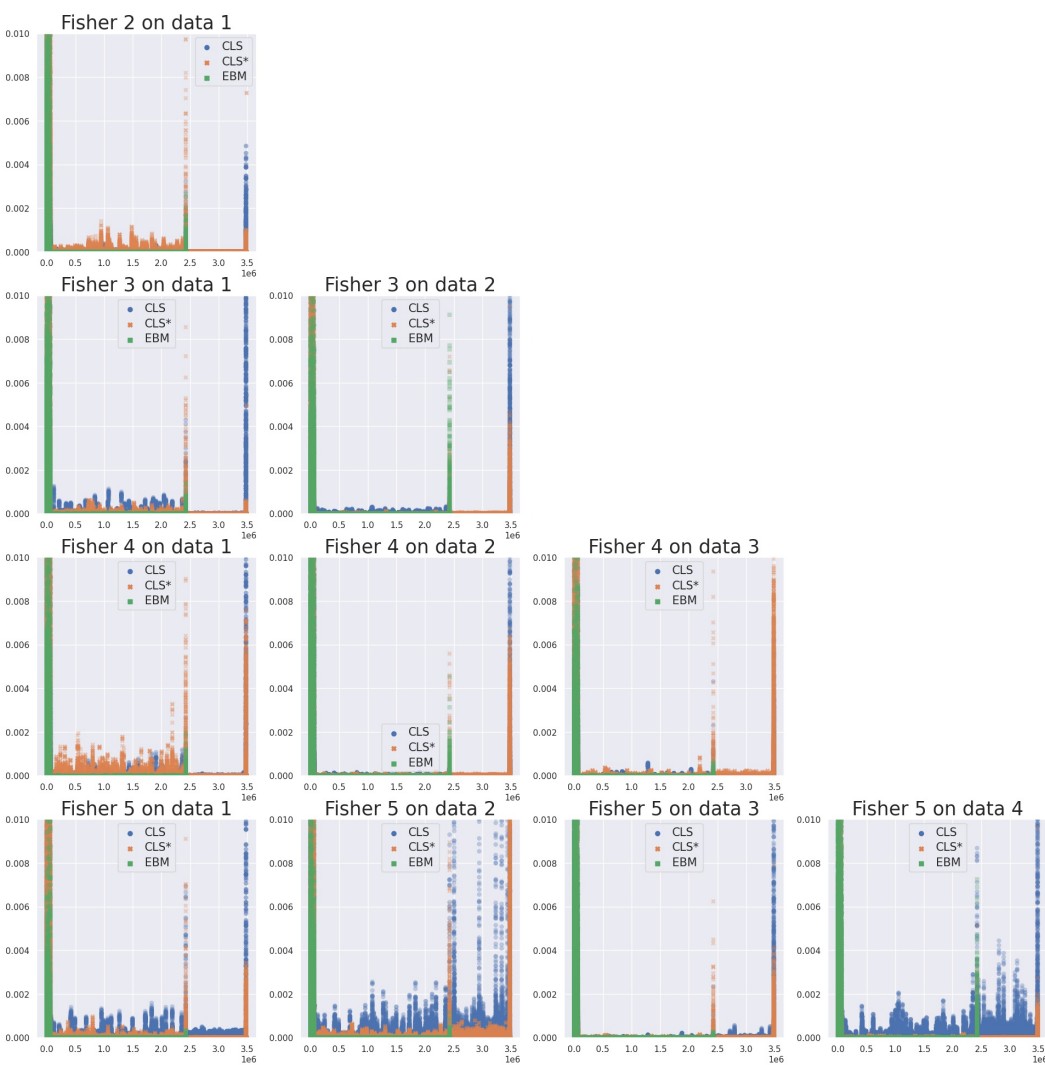

Figure 7: Parameter importance on different tasks. The x-axis represents each different parameter and y-axis is the FIM value in Equation 8. The sparser the parameters are, the fewer important parameters for previous data, which may contribute to why EBMs has less influence on previous tasks after training on new data. "Fisher 5 on data 1" means the diagonal elements of the fisher information matrix obtained by Equation 8 using the model parameters $\theta_5$ (after training on task $T_5$) and data from task $T_1$.

Table 6: Comparison of our EBM with baselines on different variants of the split CIFAR-100 protocol. The CIFAR-100 dataset is split up into 5 tasks (= 20 classes per task), 10 tasks (= 10 classes per task), 20 tasks (= 5 classes per task) or 50 tasks (= 2 classes per task). Shown is the *Class-IL* performance on the *Boundary-Aware* setting after all tasks have been learned. Each experiment is performed 10 times with different random seeds, the results are reported as the mean $\pm$ SEM over these runs.

| Method | CIFAR-100 split up into: | | | |
| | 5 tasks | 10 tasks | 20 tasks | 50 tasks |
|---|---|---|---|---|
| CLS | $14.74 \pm 0.20$ | $8.18 \pm 0.10$ | $4.46 \pm 0.03$ | $1.91 \pm 0.02$ |
| EWC | $14.78 \pm 0.21$ | $8.20 \pm 0.09$ | $4.46 \pm 0.03$ | $1.91 \pm 0.02$ |
| SI | $14.07 \pm 0.24$ | $9.24 \pm 0.22$ | $4.37 \pm 0.04$ | $1.88 \pm 0.03$ |
| LwF | $25.75 \pm 0.14$ | $10.71 \pm 0.11$ | $12.18 \pm 0.16$ | $7.68 \pm 0.16$ |
| **EBM** | $\mathbf{34.88 \pm 0.14}$ | $\mathbf{30.28 \pm 0.28}$ | $\mathbf{25.04 \pm 0.33}$ | $\mathbf{13.60 \pm 0.50}$ |

## B    DERIVATION OF LOSS GRADIENT

The derivation of the loss gradient in Equation 5 is

$$
\begin{aligned}
\frac{\partial \mathcal{L}_{\text{ML}}}{\partial \theta} &= \mathbb{E}_{\mathbf{x} \sim T} \left( \frac{\partial E_\theta(\mathbf{x}, \mathbf{y}^+)}{\partial \theta} + \frac{\partial \left[ \log \sum_{\mathbf{y}_i \in \{\mathcal{N}, \mathbf{y}^+\}} \exp(-E_\theta(\mathbf{x}, \mathbf{y}_i)) \right]}{\partial \theta} \right) \\
&= \mathbb{E}_{\mathbf{x} \sim T} \left( \frac{\partial E_\theta(\mathbf{x}, \mathbf{y}^+)}{\partial \theta} + \frac{1}{\sum_{\mathbf{y}_i \in \{\mathcal{N}, \mathbf{y}^+\}} \exp(-E_\theta(\mathbf{x}, \mathbf{y}_i))} \frac{\partial \left[ \sum_{\mathbf{y}_i \in \{\mathcal{N}, \mathbf{y}^+\}} \exp(-E_\theta(\mathbf{x}, \mathbf{y}_i)) \right]}{\partial \theta} \right) \\
&= \mathbb{E}_{\mathbf{x} \sim T} \left( \frac{\partial E_\theta(\mathbf{x}, \mathbf{y}^+)}{\partial \theta} + \frac{1}{\sum_{\mathbf{y}_i \in \{\mathcal{N}, \mathbf{y}^+\}} \exp(-E_\theta(\mathbf{x}, \mathbf{y}_i))} \sum_{\mathbf{y}_i \in \{\mathcal{N}, \mathbf{y}^+\}} \frac{\partial \left[ \exp(-E_\theta(\mathbf{x}, \mathbf{y}_i)) \right]}{\partial \theta} \right) \\
&= \mathbb{E}_{\mathbf{x} \sim T} \left( \frac{\partial E_\theta(\mathbf{x}, \mathbf{y}^+)}{\partial \theta} - \frac{1}{\sum_{\mathbf{y}_i \in \{\mathcal{N}, \mathbf{y}^+\}} \exp(-E_\theta(\mathbf{x}, \mathbf{y}_i))} \sum_{\mathbf{y}_i \in \{\mathcal{N}, \mathbf{y}^+\}} \exp(-E_\theta(\mathbf{x}, \mathbf{y}_i)) \frac{\partial E_\theta(\mathbf{x}, \mathbf{y}_i)}{\partial \theta} \right) \\
&= \mathbb{E}_{\mathbf{x} \sim T} \left( \frac{\partial E_\theta(\mathbf{x}, \mathbf{y}^+)}{\partial \theta} - \sum_{\mathbf{y}_i \in \{\mathcal{N}, \mathbf{y}^+\}} \frac{1}{\sum_{\mathbf{y}_i \in \{\mathcal{N}, \mathbf{y}^+\}} \exp(-E_\theta(\mathbf{x}, \mathbf{y}_i))} \exp(-E_\theta(\mathbf{x}, \mathbf{y}_i)) \frac{\partial E_\theta(\mathbf{x}, \mathbf{y}_i)}{\partial \theta} \right) \\
&\qquad \left( \sum_{\mathbf{y}_i \in \{\mathcal{N}, \mathbf{y}^+\}} \exp(-E_\theta(\mathbf{x}, \mathbf{y}_i)) \text{ is a constant and can be moved inside the summation} \right) \\
&= \mathbb{E}_{\mathbf{x} \sim T} \left( \frac{\partial E_\theta(\mathbf{x}, \mathbf{y}^+)}{\partial \theta} - \sum_{\mathbf{y}_i \in \{\mathcal{N}, \mathbf{y}^+\}} \frac{\exp(-E_\theta(\mathbf{x}, \mathbf{y}_i))}{\sum_{\mathbf{y}_i \in \{\mathcal{N}, \mathbf{y}^+\}} \exp(-E_\theta(\mathbf{x}, \mathbf{y}_i))} \frac{\partial E_\theta(\mathbf{x}, \mathbf{y}_i)}{\partial \theta} \right) \\
&= \mathbb{E}_{\mathbf{x} \sim T} \left( \frac{\partial E_\theta(\mathbf{x}, \mathbf{y}^+)}{\partial \theta} - \sum_{\mathbf{y}_i \in \{\mathcal{N}, \mathbf{y}^+\}} p_\theta(\mathbf{y}_i | \mathbf{x}) \frac{\partial E_\theta(\mathbf{x}, \mathbf{y}_i)}{\partial \theta} \right) \\
&= \mathbb{E}_{\mathbf{x} \sim T} \left( \frac{\partial E_\theta(\mathbf{x}, \mathbf{y}^+)}{\partial \theta} - \mathbb{E}_{\mathbf{y}_i \sim \{\mathcal{N}, \mathbf{y}^+\}} \left[ \frac{\partial E_\theta(\mathbf{x}, \mathbf{y}_i)}{\partial \theta} \right] \right).
\end{aligned}
\tag{9}
$$

## C    MODEL ARCHITECTURES

Here we provide details of the model architectures used on the different datasets.

Images from the split MNIST and permuted MNIST datasets are grey-scale images. The baseline models for these datasets, similar as in (van de Ven & Tolias, 2019), consist of several fully-connected layers. For the EBMs we use similar number of parameters. The model architectures of EBMs on the split MNIST dataset and permuted MNIST dataset are the same, but have different input and output dimensions and hidden sizes. The model architectures of EBMs and baseline models on the split MNIST dataset are shown in Table 7a and Table 7b respectively. The model architectures of EBMs and baseline models on the permuted MNIST dataset are shown in Table 8a and Table 8b.

Images from the CIFAR-10 and CIFAR-100 datasets are RGB images. For CIFAR-10, we use a small convolutional network for both the baseline models and the EBMs. The model architectures of EBMs on the CIFAR-10 dataset are shown in Table 9a, Table 9b, Table 9c, Table 9d, and Table 9e. We investigate different architectures to search for the effective label conditioning on EBMs training as described in Section 5.1.5. The baseline models on CIFAR-10 are given in Table 9f.

The model architectures used on the CIFAR-100 dataset are detailed in Table 10.

Table 7: The model architectures used on split MNIST. $h = 400$.

(a) The architecture of the EBMs.

| x = FC(784, h) (x) |
| --- |
| x = ReLU (x) |
| y = Embedding (y) |
| x = x * Norm2 (y) + x |
| x = ReLU (x) |
| out = FC(h, 1) (x) |

(b) The architecture of the baseline models.

| x = FC(784, h) (x) |
| --- |
| x = ReLU (x) |
| x = FC(h, h) (x) |
| x = ReLU (x) |
| out = FC(h, 10) (x) |

Table 8: The model architectures used on permuted MNIST. $h = 1000$.

(a) The architecture of the EBMs.

| x = FC(1024, h) (x) |
| --- |
| x = ReLU (x) |
| y = Embedding (y) |
| x = x * Norm2 (y) + x |
| x = ReLU (x) |
| out = FC(h, 1) (x) |

(b) The architecture of the baseline models.

| x = FC(1024, h) (x) |
| --- |
| x = ReLU (x) |
| x = FC(h, h) (x) |
| x = ReLU (x) |
| out = FC(h, 100) (x) |

Table 9: The model architectures used on the CIFAR-10 dataset.

(a) EBM: **Beginning (V1)**

| Input: x, y |
| --- |
| y = Embedding(N, 3) (y) |
| y = Softmax(dim=-1) (y) |
| y = y * y.shape[-1] |
| x = x * y |
| x = Conv2d(3×3, 3, 32) (x) |
| x = ReLU (x) |
| x = Conv2d(3×3, 32, 32) (x) |
| x = ReLU (x) |
| x = Maxpool(2, 2) (x) |
| x = Conv2d(3×3, 32, 64) (x) |
| x = ReLU (x) |
| x = Conv2d(3×3, 64, 64) (x) |
| x = ReLU (x) |
| x = Maxpool(2, 2) (x) |
| x = FC(2304, 1024) (x) |
| x = ReLU(x) |
| out = FC(1024, 1) (x) |

(b) EBM: **Middle (V2)**

| Input: x, y |
| --- |
| x = Conv2d(3×3, 3, 32) (x) |
| x = ReLU (x) |
| x = Conv2d(3×3, 32, 32) (x) |
| x = ReLU (x) |
| y = Embedding(N, 32) (y) |
| y = Softmax(dim=-1) (y) |
| y = y * y.shape[-1] |
| x = x * y |
| x = Maxpool(2, 2) (x) |
| x = Conv2d(3×3, 32, 64) (x) |
| x = ReLU (x) |
| x = Conv2d(3×3, 64, 64) (x) |
| x = ReLU (x) |
| x = Maxpool(2, 2) (x) |
| x = FC(2304, 1024) (x) |
| x = ReLU(x) |
| out = FC(1024, 1) (x) |

(c) EBM: **Middle (V3)**

| Input: x, y |
| --- |
| x = Conv2d(3×3, 3, 32) (x) |
| x = ReLU (x) |
| x = Conv2d(3×3, 32, 32) (x) |
| x = ReLU (x) |
| x = Maxpool(2, 2) (x) |
| x = Conv2d(3×3, 32, 64) (x) |
| x = ReLU (x) |
| x = Conv2d(3×3, 64, 64) (x) |
| x = ReLU (x) |
| y = Embedding(N, 64) (y) |
| y = Softmax(dim=-1) (y) |
| y = y * y.shape[-1] |
| x = x * y |
| x = Maxpool(2, 2) (x) |
| x = FC(2304, 1024) (x) |
| x = ReLU(x) |
| out = FC(1024, 1) (x) |

(d) EBM: **End Fix Softmax (V4)**

| Input: x, y |
| --- |
| x = Conv2d(3×3, 3, 32) (x) |
| x = ReLU (x) |
| x = Conv2d(3×3, 32, 32) (x) |
| x = ReLU (x) |
| x = Maxpool(2, 2) (x) |
| x = Conv2d(3×3, 32, 64) (x) |
| x = ReLU (x) |
| x = Conv2d(3×3, 64, 64) (x) |
| x = ReLU (x) |
| x = Maxpool(2, 2) (x) |
| x = FC(2304, 1024) (x) |
| y = Random Projection (y) |
| y = Softmax(dim=-1) (y) |
| y = y * y.shape[-1] |
| x = x * y |
| out = FC(1024, 1) (x) |

(e) EBM: **End Softmax (V4)**

| Input: x, y |
| --- |
| x = Conv2d(3×3, 3, 32) (x) |
| x = ReLU (x) |
| x = Conv2d(3×3, 32, 32) (x) |
| x = ReLU (x) |
| x = Maxpool(2, 2) (x) |
| x = Conv2d(3×3, 32, 64) (x) |
| x = ReLU (x) |
| x = Conv2d(3×3, 64, 64) (x) |
| x = ReLU (x) |
| x = Maxpool(2, 2) (x) |
| x = FC(2304, 1024) (x) |
| y = Embedding(N, 1024) (y) |
| y = Softmax(dim=-1) (y) |
| y = y * y.shape[-1] |
| x = x * y |
| out = FC(1024, 1) (x) |

(f) Baseline models

| Input: x |
| --- |
| x = Conv2d(3×3, 3, 32) (x) |
| x = ReLU (x) |
| x = Conv2d(3×3, 32, 32) (x) |
| x = ReLU (x) |
| x = Maxpool(2, 2) (x) |
| x = Conv2d(3×3, 32, 64) (x) |
| x = ReLU (x) |
| x = Conv2d(3×3, 64, 64) (x) |
| x = ReLU (x) |
| x = Maxpool(2, 2) (x) |
| x = FC(2304, 1024) (x) |
| x = ReLU (x) |
| x = FC(1024, 1024) (x) |
| x = ReLU (x) |
| out = FC(1024, 10) (x) |

Table 10: The model architectures used on the CIFAR-100 dataset. Following van de Ven et al. (2020), for all models the convolutational layers were pre-trained on CIFAR-10. The 'BinaryMask'-operation fully gates a randomly selected subset of $X\%$ of the nodes, with a different subset for each $y$. Hyperparameter $X$ was set using a gridsearch.

(a) The architecture of the EBMs.

| Input: x, y |
| --- |
| x = Conv2d(3×3, 3, 16) (x) |
| x = BatchNorm (x) |
| x = ReLU (x) |
| x = Conv2d(3×3, 16, 32) (x) |
| x = BatchNorm (x) |
| x = ReLU (x) |
| x = Conv2d(3×3, 32, 64) (x) |
| x = BatchNorm (x) |
| x = ReLU (x) |
| x = Conv2d(3×3, 64, 128) (x) |
| x = BatchNorm (x) |
| x = ReLU (x) |
| x = Conv2d(3×3, 128, 256) (x) |
| x = FC(1024, 2000) (x) |
| x = ReLU (x) |
| x = BinaryMask (x, y) |
| x = FC(2000, 2000) (x) |
| x = ReLU (x) |
| x = BinaryMask (x, y) |
| out = FC(2000, 1) (x) |

(b) The architecture of the baseline models.

| Input: x |
| --- |
| x = Conv2d(3×3, 3, 16) (x) |
| x = BatchNorm (x) |
| x = ReLU (x) |
| x = Conv2d(3×3, 16, 32) (x) |
| x = BatchNorm (x) |
| x = ReLU (x) |
| x = Conv2d(3×3, 32, 64) (x) |
| x = BatchNorm (x) |
| x = ReLU (x) |
| x = Conv2d(3×3, 64, 128) (x) |
| x = BatchNorm (x) |
| x = ReLU (x) |
| x = Conv2d(3×3, 128, 256) (x) |
| x = FC(1024, 2000) (x) |
| x = ReLU (x) |
| x = FC(2000, 2000) (x) |
| x = ReLU (x) |
| out = FC(2000, 100) (x) |

