# OpenReview forum: "Energy-Based Models for Continual Learning"
_ICLR.cc/2021/Conference — Reject_

### Official Review · AnonReviewer2 · 2020-10-26
**This paper proposes an energy-based model for continual learning, which seems to be new. However, I have several concerns about the paper, which are described in the detailed comments.**

**Rating:** 4
**Confidence:** 4

**Review:**

The writing of the paper needs improvements. I am still unsure how the paper learns each new task. You talked about batches but never talked about how each new task is learned specifically. It creates doubts in my mind. E.g., does each batch contain some examples from old tasks? Is this training like for multi-task learning?

You wrote “In the continual learning setting, we assume classes in Y_B are uniformly distributed in every new batch.” Is Y_B fixed for each task or each batch?

You need a negative class in each batch. This means that your algorithm cannot work in the scenario where a task has one class only. Most of existing techniques can handle this case although they use 2 or more classes in a task in their experiments. I think this is a serious limitation. It is only suitable for Task-IL, which is an easier problem to solve.

I think the claim that existing techniques need to fix the number of classes beforehand is not correct. I know hat most of them fix the number in their code or experiments, but I don’t see why they cannot use a large number or dynamically add new class heads when needed, say, using cross-entropy as the loss function.

The experiments are not well described, and baselines are old. Many newer baselines should be included, e.g.,

Learning a Unified Classifier Incrementally via Rebalancing. CVPR 2019.
Overcoming catastrophic forgetting for continual learning via model adaptation. ICLR, 2019.
Random path selection for continual learning. NeurIPS 2019
Continuous learning of context-dependent processing in neural networks. Nature Machine Intelligence, 2019
Large scale incremental learning. CVPR 2019

For each dataset, you used one setting for tasks only, e.g., CIFAR100, 10 tasks. More than one setting should be tried to show the generality of the approach.

---

> ### Author Response · Authors · 2020-11-18
> **Author Response to AnonReviewer2 (1 of 2)**
>
> We thank the reviewer for their thorough and insightful review. In response to this review, we have improved the writing and the structure of our paper, extended our discussion of the continual learning literature, described how the selection of negative samples could be generalized, and added additional experiments to test the generality of our approach.
>
> Q1: Does each batch contain some examples from old tasks? Is this training like for multi-task learning?
>
> We apologize that the original paper was not clear on this. It is indeed important to point out that the batches that our models are trained on do not contain any examples from old tasks, and that we thus do not do multi-task training. In fact, none of the methods that we consider store data from old tasks or use any form of replay (see also below). We have clarified both of these points in the revised paper (e.g., see Sec 5.1.2).
>
> Q2: You wrote, “we assume classes in Y_B are uniformly distributed in every new batch.” Is Y_B fixed for each task or each batch?
>
> We apologize that our statement caused confusion. In the revised paper we have improved the writing of this part of the paper (see Sec 4.1). In particular, we now made a distinction between the set of class labels that appears in the current batch (denoted by Y_B) and the set of class labels that are used as negative samples (denoted by N). Note that in our CL experiments, typically the set of negative samples was sampled from the labels in the current batch (i.e., N was sampled from Y_B).
> We further note that the data in each batch is randomly sampled from the current task’s training data, so Y_B can be different for each new batch and it will certainly be different for each new task.
>
> Q3: Your algorithm cannot work in the scenario where a task has one class only. Most existing techniques can handle this case. Your EBM is only suitable for Task-IL.
>
> We thank the reviewer for this insightful comment. We understand that from our original formulation it seemed like that our EBMs cannot deal with the setting where each task only has one class. However, we want to highlight a few things:
>
> (1) Firstly, although in our initial submission we indeed defined our EBMs as sampling their negative sample(s) from the current batch, this is not actually necessary. It is possible to use different strategies for the selection of the negative sample(s), as we explore in Table 2. For example, one might instead use whichever other class was seen most recently as a negative sample, or one could sample from all other classes seen so far. Another solution might be to always use a fake label representing “not exist” as negative samples. In the revised paper (Sec 4.1), we now discuss these alternative strategies for the selection of negative samples.
>
> (2) Secondly, the statement that most existing techniques can handle the case with only one class per task is somewhat overstated. We agree that most methods that store data or that use replay are capable of handling this case, but we are not aware of any existing non-replay methods that could successfully do this. (Technically, standard classifiers or regularization-based methods could be trained on the class-incremental scenario with a single class per task, but without some form of replay these methods will do very badly.) Moreover, if our method were to be combined with replay (i.e., adding examples from old tasks to the current batch), then our method -- even in the way we had formulated it in the initial submission -- will also be able to handle this case (as then there will be multiple labels in each batch).
>
> (3) Finally, we want to point out that our method is certainly suitable for the Class-IL scenario (all of the experiments reported in our paper are for the Class-IL scenario). The sampling from the current batch is only for TRAINING, during testing we feed in all possible labels and select the label with the lowest energy as the prediction (see Eq 7). Although we do not doubt that the reviewer had appreciated this already, we felt we had to point this out just in case, because the final sentence of the above comment is somewhat ambiguous.
>
>
> Q4: The claim that existing techniques need to fix the number of classes beforehand is not correct.
>
> Thank you for this feedback; we now realize that we overstated this claim in our original submission. In the revised paper we have removed our claim that existing techniques are not able to dynamically adding new classes. However, we still believe EBMs have fundamental benefits than the cross-entropy loss when it comes to dynamically add new classes. EBMs do not need to modify the model architecture or resize the network when adding new classes while the cross-entropy loss based methods have to replace the final classifier layer. We now more factually point out that there is a difference between how our EBM deals with new adding new classes versus how a standard classifier does this (see Section 4.2).

---

> > ### Author Response · Authors · 2020-11-18
> > **Author Response to AnonReviewer2 (2 of 2)**
> >
> > Q5: The experiments are not well described, and the baselines are old. Many newer baselines should be included.
> >
> > Thanks for these suggestions. In our revised paper we now discuss these papers in our overview of the existing continual learning literature in (Sec 2). Importantly, however, for varying reasons these papers are not suitable as baselines in our comparison:
> >
> > (1) Two of the suggested papers (“Overcoming catastrophic forgetting for continual learning via model adaptation [ICLR, 2019]” and “Continuous learning of context-dependent processing in neural networks [Nat Mach Intel, 2019]”) propose methods that are only suitable for task-incremental learning, and not for class-incremental learning.
> >
> > (2) The remaining three suggested papers (“Learning a Unified Classifier Incrementally via Rebalancing [CVPR, 2019]”, “Random path selection for continual learning [NeurIPS, 2019]” and “Large scale incremental learning [CVPR, 2019]”) address the class-incremental learning problem, but they do so by storing a subset of data from previously seen classes. In our paper, we try to do class-incremental learning WITHOUT storing ANY data.
> >
> > We believe that this set of papers highlights that the problem addressed in our paper -- class-incremental learning without using replay or stored data -- is truly a very challenging problem, and we believe that our results are an important first step towards tackling this problem.
> >
> > Finally, we have also added improved and extended descriptions of our experiments and training details (see Sections 2.1, 5.1.2 & 5.2.2, and Appendix C).
> >
> > Q6: For each dataset, you used one setting for tasks only, e.g., CIFAR100, 10 tasks. More than one setting should be tried to show the generality of the approach.
> >
> > We agree that it is important to test a method on a wide variety of settings, and it is a good suggestion to repeat some of our protocols with different numbers of tasks. In the revised paper, we now perform the Split CIFAR-100 experiments with 5, 10, 20, and 50 tasks (see Appendix A.6 and Table 6). On all of these settings we find comparable results, with our EBMs outperforming the baselines.

---

> ### Author Response · Authors · 2020-11-24
> **Re-evaluation Based on Rebuttal and Revision**
>
> Dear AnonReviewer2,
>
> Thank you very much for your thorough and insightful review. We spent a large amount of work answering the questions initially requested. We would appreciate it if you could take a look at the revised version and re-evaluate our work.
>
> Many thanks!
>
> Paper Authors

---

### Official Review · AnonReviewer3 · 2020-10-27
**Review 3**

**Rating:** 6
**Confidence:** 3

**Review:**

Summary: This work shows that energy-based models (EBMs)  are a promising model class for continual learning problems. According to the experiments, EBMs outperform the baseline methods by several continual learning benchmarks.

+ves:
1. It is interesting to see that energy-based models are introduced for the classification continual learning problems. In the paper, the authors show that EBMs achieve a significant improvement on four standard CL benchmarks. It is really surprising that the improvements are so large.

2. Some analysis, for example,  “energy landscape” and “Interference with past data.” seems interesting. It shows benefit of energy-based model for continual learning. Both of them indicate that y EBMs suffer less from catastrophic forgetting.


Concerns
1. It seems like the gradient computation of equation (5) is not corresponding to equation (4).

2. It is nice to see the improvement of EBMS. In this work, a difference architecture is usually in EBMS. The architecture is different from the baseline. It is nice that EMBS has a more flexible architecture to score the input x and output y. However, the improvement of your work is from architecture or from the training. It is better to do a clear claim. If the benefit is from the architecture, do you treat it as a black box? Or it is from the training objective?  It is possible to show some learned structure in your formulation. The architecture of EBM seems large.


Questions during the rebuttal period:
Please address and clarify the cons above:

1. The experiment setting detail is unclear. The training details of the proposed method and baseline are unclear.  And the baseline detail is a little confusing to me. In section 5.1.2, it says “ All the baselines and EBMs are based on the same model architecture”. It seems the architecture of EBMs is different.

2. Some related work on energy-based model: multiple label classification[1] , sequence labeling [2] and machine translation [3]
[1] Structured Prediction Energy Networks, ICML 2016
[2] Benchmarking Approximate Inference Methods for Neural Structured Prediction. NAACL 2019
[3] ENGINE: Energy-Based Inference Networks for Non-Autoregressive Machine Translation
ACL 2020

---

> ### Author Response · Authors · 2020-11-18
> **Author Response to AnonReviewer3**
>
> Thanks for the feedback. We agree the mentioned papers are related to our work and we now discuss the mentioned papers in Sec 3.2 and Sec 4.1 in the rebuttal version.
>
> Q1: It seems like the gradient computation of equation (5) is not corresponding to equation (4).
>
> We think Eqn(5) is correct, but we do apologize for not providing proof as the derivation is indeed not very straight-forward. In the revised paper we have now added the derivation of the loss gradient in Eqn(9) in Appendix B.
>
>
> Q2:  (1) The architecture of EBMs is different from baselines. The improvement of EBMs is from architecture or from the training? If the benefit is from the architecture, do you treat it as a black box? Or is it from the training objective?
> (2) The architecture of EBM seems large.
>
> (1) The improvement of our EBMs is from BOTH the architecture (i.e. label conditioning) and the training objective (i.e. negative sampling). We reorganized Section 5.1.4 and 5.1.5 in the rebuttal version to make this more clear.
>
> First, we show in Section 5.1.4 that the proposed EBM training objective is important. In table 2, we show for CIFAR-100 the results of 3 different training objectives: “All Neg Seen (V4)”, “All Neg Batch (V4)”, and “1 Neg Batch (V4)”, using the same model architecture. The result of “1 Neg Batch (V4)” is the best.
>
> Second, in table 3, we replace the training objective of baseline models using our EBM training objective. Our EBM training objective can improve the performance of baselines. However, even using our training objective, the baseline approaches are still worse than our EBM, which indicates that the EBM architecture is also important.
>
> Thirdly, in Section 5.1.5 we ask what type of EBM architecture is most helpful. Table 2 reports the results of 9 different architectures on CIFAR-10: “Beginning (V1)”, “Middle(V2)”, “Middle(V3)”, “End Fix (V4)”, “End Fix Norm2 (V4)”, “End Fix Softmax (V4)”, “End (V4)”, “End Norm2 (V4)”, and “End Softmax (V4)”, using the same training objective “1 Neg Batch”. We find that the architecture “End Softmax (V4)” gives the best results.
>
> (2) Our EBMs are actually slightly smaller than the baselines. This is because the EBMs have one fewer FC layer than the baselines (see Appendix C). On split MNIST, EBMs have 400*400-400=159600 fewer parameters than the baselines; on permuted MNIST, EBMs have 1000*1000-1000=999000 fewer parameters; and on CIFAR-10 and CIFAR-100, EBMs have 1024*1024-1024=1047552 fewer parameters.
>
>
> Q3: The experiment setting detail is unclear. The training details of the proposed method and baseline are unclear.
>
> We apologize that the original paper was not clear on this. In the revised paper we have added more descriptions of the experiment setting details in Sec 2.1.
> The boundary-aware setting is the standard continual learning setting with clear task boundaries and tasks are given sequentially. “Three scenarios for continual learning Van de Ven & Tolias, 2019”, has a clear illustration in their Fig1 and Tab2.
> The boundary-agnostic setting has no task boundaries. “Zeno et al., 2018 Task Agnostic Continual Learning Using Online Variational Bayes” show this setting in their Fig2. Number of samples from each task in each batch is a random variable drawn from a distribution over tasks, and this distribution changes over time.
>
> In the revised paper, we have improved and extended the description of the training details (see Sections 5.1.2 & 5.2.2, and Appendix C).
>
>
> Q4: The baseline detail is a little confusing to me. In section 5.1.2, it says “ All the baselines and EBMs are based on the same model architecture”. It seems the architecture of EBMs is different.
>
> We apologize that our statement caused confusion. What we meant to say was that we used "similar" or "comparable" architectures and that they are based on the same backbone architectures.
>
> For example, for CIFAR-10, the baselines have architecture:
> x->Conv2d->ReLU->Conv2d->ReLU->Maxpool->Conv2d->ReLU->Conv2d->ReLU->Maxpool->FC->ReLU->FC->ReLU-->FC(1024,N)
>
> Our EBM variant “End Softmax (V4)”, which is the variant that is used to compared with baselines in Table 1, Table 3 and Table 4, has architecture:   x->Conv2d->ReLU->Conv2d->ReLU->Maxpool->Conv2d->ReLU->Conv2d->ReLU->Maxpool->FC
>
> y->FC(N,1024)->Softmax->*x->FC(1024, 1)
>
> The backbones of EBM and baselines are the same, i.e. Conv2d->ReLU->Conv2d->ReLU->Maxpool->Conv2d->ReLU->Conv2d->ReLU->Maxpool->FC. Only the other 1 or 2 layers are slightly different.
>
> In the revised paper, we now more clearly describe the architectures of all models in Appendix C (Tables 7, 8, 9 & 10).

---

### Official Review · AnonReviewer4 · 2020-10-29
**Insufficient novelty + relevant reference missing**

**Rating:** 5
**Confidence:** 4

**Review:**

==========================

before revision

==========================

Review: Motivated by the effectiveness and naturalness of energy-based models, this paper proposes to use energy-based learning framework for continual learning. Empirical studies are performed to validate the proposed strategy on several continual learning benchmarks.

Strength:
+ This paper applies EBMs to the task of continual learning, which is interesting and relevant to ICLR conference.
+ The paper is well written and easy to follow.
+ The paper is technically sound, since the formulation of the EBMs are well derived by other prior works.


Concerns:
+ The contribution of the paper is insufficient for publication. The energy-based learning framework for discriminative purpose has been developed for a long time, even though recently researchers in the field machine learning are enthusiastic about developing energy-based models for data generation.
+ The underlying theory of the proposed method is developed by other papers, the only contribution of this paper is to apply the EBM to the continual learning, which is quite straightforward.
+ Missing key reference about EBM for discriminative learning. The core of this paper is mainly based on the finding of the transition between discriminative EBM and generative EBM, which is originally presented in reference [a]. The current paper misses to discuss and cite this paper.
+ missing relevant reference about generative EBMs in related work. Even though this paper is not directly related to EBMs for data generation, but it DID discuss the development of it in its paper. The current related work about EBMs for generative purpose is incomplete in the sense that it skipped some pioneering works and important application with EBMs. For examples,  [1] is the first paper to use ConvNet-parameterized EBMs with Langevin for image generation.  Training EBMs with assisting networks can be found in [2] and [3].  Also, writing a section of comprehensive related works about energy-based learning is not necessary but encouraging.

references
+ [1] A Theory of Generative ConvNet (ICML 2016)
+ [2] Cooperative learning of descriptor and generator networks. IEEE Transactions on Pattern Analysis and Machine Intelligence (PAMI 2018).
+ [3] Divergence triangle for joint training of generator model, energy-based model, and inference model. (CVPR 2019)

===================================

After a revision

===================================
Thank you for your efforts to revise the paper. The revised parts about related work look good to me. I agree on that citing all those EBM application papers is not necessary. But doing so can provide a comprehensive and complete development of  the DeepNet-EBM. Again, this is not required and it will not affect the rating.

I also acknowledge the existing contributions in the current paper and admit that such a direction is promising, but I still feel that the current paper doesn't fully explore this area with more solid experiments. Thus, the whole contribution is quite marginal. By taking into account all these concerns, I will change my rating from 4 to 5.

---

> ### Author Response · Authors · 2020-11-18
> **Author Response to AnonReviewer4**
>
> Q1: Missing key reference about EBM for discriminative learning.
> Q2: Missing relevant references for generative EBMs.
> Q3: The only contribution of this paper is to apply the EBM to continual learning, which is quite straightforward.
>
> We thank the reviewer for their comments and suggestions.
>
> We agree that in our original submission the discussion of how our formulation of EBMs for classification relates to previous work using EBMs for discriminative tasks fell short. In the revised paper, we now clearly discuss how our paper relates to this line of previous work (see Section 3.2 and Section 4.1).
>
> We thank the reviewer for their suggestion on how to improve our discussion of the use of EBMs for generative purposes; we have done so in the revised paper (see Section 3.2). However, given that the main topic of our paper is the use of EBMs for classification-based continual learning, we believe that citing all five papers suggested by the reviewer would be unproportionate.
>
> The main concern of this reviewer is that the application of EBMs to continual learning is “quite straight-forward”. We respectfully disagree with this.
>
> Our paper addresses two fundamental problems in continual learning: (1) How can a neural network do class-incremental learning without relying on stored data or replay? (2) How can a neural network incrementally learn new information without relying on explicit task boundaries? If anything, we believe that demonstrating that a *relatively simple* framework can successfully address these problems should be considered a strength.
>
> Moreover, we show that naively applying EBMs to continual learning will not necessarily work. Our experiments reported in Table 2 demonstrate that both the specification of the energy training objective (i.e., the way the negative samples are selected) and the way the energy network is set up (i.e., the label conditioning architecture) substantially influence how successful catastrophic forgetting is prevented in an EBM.
>
> In addition, our paper provides deep analysis and visualizations from different perspectives to verify why EBMs help prevent catastrophic forgetting. We give a quantification of interference with past data in Sec 5.1.6, a comparison of model capacity in A.3 and an analysis of parameter importance in A.4; and we provide visualizations of the energy landscapes (Fig 2), the predicted class distributions (Fig 3), the testing curves (Fig 4) and the confusion matrix (Fig 5).

---

> ### Author Response · Authors · 2020-11-24
> **Clarifications on Experimental Evaluation**
>
> Dear Reviewer,
>
> Thank you very much for your reply. We are encouraged that you think our EBM for continual learning is promising.
>
> In response to your comment ", I still feel that the current paper doesn't fully explore this area with more solid experiments", we have listed the experiments that we have done:
>
> 1. We report results on 4 datasets, including Split MNIST, Permuted MNIST, CIFAR-10, and CIFAR-100, in Tables 1 and 4. To our knowledge, most existing continual learning works only show their results on 2 to 3 datasets.
>
> 2. We show that our model performs well on the boundary-aware setting, a commonly used evaluation setting for existing continual learning approaches, on different datasets in Table 1.
>
> 3. We further evaluate a more challenging boundary agnostic setting in Table 4 where many continual learning methods don't hold, but our EBMs still show good performance.
>
> 4. We investigate different training objectives and model architectures in Table 2 of the proposed model.
>
> 5. We compare the baselines using our training objective and their original one and prove that our training objective can also improve the performance of baseline methods in Table 3.
>
> 6. In the rebuttal version, we add new experiments to compare our EBM with baselines on different variants of the split CIFAR-100 protocol, to further test the scalability of our results. In Table 6, the CIFAR-100 dataset is split up into 5, 10, 20, or 50 tasks. For all settings, we find that our EBM substantially outperforms the baselines.
>
> 7. We provide deep analysis and visualizations from different perspectives to verify why EBMs help prevent catastrophic forgetting.
>
>     (a). We give a quantification of interference with past data in Sec 5.1.6. We show EBMs have less interference with past data when learning new data.
>
>     (b). We give a comparison between our EBMs and baselines of their model capacity in A.3. We show EBMs have a larger model capacity that provides another reason for why EBMs suffer less from catastrophic forgetting than standard classifiers.
>
>     (c). We provide an analysis of parameter importance in A.4. We show EBMs have fewer important parameters for previous data and thus updating parameters for the new task will have a less negative impact on old tasks.
>
>     (d). We provide visualizations of the energy landscapes in Fig 2 to show how the energy landscapes change during the training process. The energy landscapes provide more direct evidence for why EBMs are better for continual learning.
>
>     (e). We show the predicted class distributions in Fig 3 to show how the predicted classes change during the training process. EBMs can predict classes for all seen classes while the baseline method can only predict classes from the current task, indicating forgetting.
>
>     (f). We show the testing curves in Fig 4. The curve on EBMs drops slower than baselines, implying the proposed energy objective can mitigate the catastrophic forgetting problem.
>
>     (g). We show the confusion matrix in Fig 5. EBMs have high values along the diagonal that means the predicted results matching the ground truth labels for all the sequentially learned tasks. This demonstrates that EBMs are better at learning new tasks without catastrophically forgetting old tasks.
>
> We think we have shown many experiments that are necessary for continual learning in the main paper and the appendix.
>
> Please let us know if there are any other experiments you would like us to run.
>
>
> Many thanks!
>
> Paper Authors

---

### Official Review · AnonReviewer1 · 2020-11-01
**Maximum likelihood training results in catastrophic forgetting**

**Rating:** 6
**Confidence:** 4

**Review:**

This paper explores the usage of EBMs in continual learning for classification. Although the application of EBMs in continual learning is novel, the general idea is a special case of the usage of EBMs for structured prediction, which has been widely studied. For instance, multi-class classification can be considered as a special version of multi-label classification, which has been studied in Belanger and McCallum (2016) and a set of follow-up works. The main difference here is that multi-class classification is a simpler problem, and all possible classes can be enumerated in O(N), but in multi-label classification, more complicated inference such as gradient-descent based approaches must be used.
The contrastive training can be seen as a special case of margin-based training (Belanger and McCallum, 2016; Rooshenas et al. 2019), where the margin is infinity.
I believe the works in using EBMs for structured prediction must be cited here as they are closely related.

The authors also explored the effect of ML training on interference with past data and showed that using single sample ML approximation can significantly alleviate the catastrophic forgetting problem.
I believe that this is an interesting observation.

Typo: "current bath" in Section 5.1.4

Belanger and McCallum (2016), Structured Prediction Energy Networks.
Gygli et al. (2017), Deep Value Networks Learn to Evaluate and Iteratively Refine Structured Outputs.
Rooshenas et al. (2019), Search-Guided, Lightly-supervised Training of Structured Prediction Energy Networks

---

> ### Author Response · Authors · 2020-11-18
> **Author Response to AnonReviewer1**
>
> Q1: The general idea is a special case of the usage of EBMs for structured prediction. Multi-class classification can be considered as a special version of multi-label classification. The contrastive training can be seen as a special case of margin-based training. I believe the works in using EBMs for structured prediction must be cited here as they are closely related.
>
> Thanks for the feedback. We agree structured prediction is related to our work and we now discuss the mentioned papers in Sec 3.2 and Sec 4.1 in the rebuttal version.
>
> 1) Different from structured prediction that can access data all at once, continual learning is trained on sequential tasks. Ideally, CL will not access previous data while training on new data and thus the biggest challenge is to prevent catastrophic forgetting. The main focus of this paper is to mitigate the catastrophic forgetting in CL. Our approach proposes a one-negative sampling protocol specialized for continual learning and samples negative samples based on the task on hand, which is sequentially given. In contrast, SPENs utilizes a relaxed continuous optimization protocol to find multi-class labels for tasks on hand which are jointly given.
>
> 2) While both margin-based training and our own objective (Negative Log-Likelihood Loss) minimize the energy of real samples and maximize the energy of fake samples, we believe their functional forms are different. Our objective seeks to minimize the negative log-likelihood of the real sample and partition function. In contrast, the margin-based loss seeks to maximize the margin difference between the real sample and selected negative samples. Both objectives are discussed equally in Lecun’s tutorial: [A Tutorial on Energy-Based Learning in Section 2.2.3 and Section 2.2.4] and have both been used in energy-based learning. In this paper, we show that the Negative Log-Likelihood Loss can be helpful for continual learning.
>
> Q2: Typo: "current bath" in Section 5.1.4
>
> We have fixed the typo.

---

### Author Response · Authors · 2020-11-20
**General Response**

Dear Reviewers,

Thank you very much for your thorough and insightful review. We are encouraged that you think energy-based models (EBMs) are a promising model class for continual learning problems. We have provided the feedback and updated the paper.

We have improved the writing and the structure of our paper, extended our discussion of related works, added more descriptions of the experiment setting and training details,  and added additional experiments to test the generality of our approach. We provide more details in the direct responses to the reviewers.

Many thanks!

Paper Authors

---

### Author Response · Authors · 2020-11-22
**Re-evaluation Based on Rebuttal and Revision**

Dear Reviewers,

Thank you very much for your thorough and insightful review. We are encouraged that you think energy-based models (EBMs) are a promising model class for continual learning problems.

We spent a large amount of work answering the questions initially requested. We would appreciate it if you could take a look at the revised version and re-evaluate our work.

Many thanks!

Paper Authors

---

### Decision · Program_Chairs · 2021-01-07
**Final Decision**

**Decision:**

Reject

**Comment:**

There were opinions on both sides of this paper from the reviewers.  Reviewers were excited by the novel application of energy-based models (EBMs) to continual learning and the resulting performance gains, but were concerned by the more direct application of EBMs (which has been explored in other work, and here adapted to the continual learning setting, so its contribution is marginal) and with the depth of the evaluation, which they thought could be pushed farther. Overall, the reviewers agreed that this paper could benefit from another round of revisions to strengthen its contribution, incorporating many of the excellent points made by the authors in their responses.